

# Theorems for the lightcone bootstrap

**Balt C. van Rees**

CPHT, CNRS, Ecole Polytechnique, Institut Polytechnique de Paris, Palaiseau, France

## Abstract

Consider a conformally covariant four-point function of identical scalar operators with a discrete spectrum, a twist gap, and compatible with the unitarity conditions. We give a mathematical proof confirming that the spectrum and OPE coefficients at large spin and fixed twist always become that of a generalized free field theory.


## 1 Introduction

Independently of the exact axiomatic framework one wishes to adopt, it is commonly accepted that the correlation functions of local operators in conformal field theories in Euclidean $\mathbb{R}^d$ are mathematically well-defined. It is therefore worthwhile to rigorously prove the wide variety of claims regarding their properties, like the lightcone bootstrap [1, 2], the Lorentzian inversion formula [3], the conformal dispersion relation [4], features of Regge trajectories [5, 6], ANEC positivity and other properties of lightray operators [7, 8], the Polyakov-Regge block decomposition [9], the existence of Mellin amplitudes [10, 11], and dispersive functionals that

are useful for the flat-space limit of theories in AdS [12, 13]. The feasibility of such an approach is demonstrated, for example, by prior rigorous results on OPE convergence [14, 15], the Wightman functions [16, 17], and the leading Regge trajectory [18].

In this work we prove the essential claim of the lightcone bootstrap formulated in 2012 [1, 2], see [17] for a discussion. We in particular extend the results of [18] by rigorously demonstrating the existence of all the subleading double twist Regge trajectories. A corollary of our analysis is an improved understanding of the lightcone limit on the second sheet, which will be necessary for proving many of the claims in the papers listed above.

Concretely we will show that the spectrum and OPE coefficients in a CFT four-point function of identical scalar operators become those of a generalized free field theory at large spin and fixed twist, assuming unitarity, a twist gap, and a discrete spectrum. The idea is of course to approach the lightcone by taking $\bar{z} \to 1$. We can then employ a Tauberian theorem to infer convergence of the $z$-dependent part of the correlation function to the $t$-channel identity block, which is $(1-z)^{-\Delta}$ in our conventions. Previous non-rigorous approaches then proceeded to series expand around $z = 0$, but this is not justified because the Tauberian theorem only works pointwise for $0 < z < 1$. Vitali's theorem in complex analysis however tells us that the actual domain of convergence is much larger and extends to all the other sheets that are reachable by going around $z = 0$. We then write $z = e^{x+iy}$ with $x < 0$, Fourier transform with respect to $y \in \mathbb{R}$ to extract the spectrum, and show that it converges distributionally to that of a generalized free field.

## 2 Definitions

We will be concerned with functions $G(z, \bar{z})$ of the form

$$G(z, \bar{z}) = z^{-\Delta} \bar{z}^{-\Delta} \sum_{k=0}^{\infty} \sum_{j=0}^{l_k} c_{k,j} z^{\frac{E_k - l_k}{2} + j} \bar{z}^{\frac{E_k + l_k}{2} - j}, \tag{1}$$

with $z$ and $\bar{z}$ independent complex variables, real symmetric coefficients $c_{k,j} = c_{k,l_k-j} \geq 0$ and real exponents $E_k \geq 0$ and $\Delta > 0$.[1] By the CFT axioms, we suppose that the double sum is absolutely convergent in the punctured polydisk $0 < |z|, |\bar{z}| < 1$. Then $G(z, \bar{z})$ is locally analytic in this domain, with branch cuts that extend along the negative real axes.

We will assume that the sequence $\{E_k\}$ is *discrete* in the sense that $\lim_{k\to\infty} E_k = \infty$. The term with $k = 0$ is fixed to be the *identity contribution* with $E_0 = 0$, $\ell_0 = 0$, and $c_{0,0} = 1$. We will also assume a *twist gap* which means that $\inf_{k>0}(E_k - l_k) > 0$.

Finally we will assume crossing symmetry:

$$G(z, \bar{z}) = G(1 - z, 1 - \bar{z}). \tag{2}$$

## 3 Asymptotics

We will explore the analytic continuation of $G(z, \bar{z})$ through the branch cuts along the negative real $z$ axis. So we write $z = e^w$ and let $g(w, \bar{z})$ be the analytic continuation of $G(e^w, \bar{z})$ from the negative real $w$ axis to the entire left half plane with $\text{Re}(w) < 0$.

Let us re-order the sums in $g(w, \bar{z})$ as:

$$g(w, \bar{z}) = \sum_n f_n(w) \bar{z}^{\bar{h}_n - \Delta}, \qquad f_n(w) := \sum_m d_{m,n} e^{(h_{m,n} - \Delta)w}. \tag{3}$$

---

[1] The positivity of the $c_{k,j}$ follows from reflection positivity, see appendix A.2 of [1] or the arguments in [19, 20].

Here the index $n$ labels the sequence $\{\bar{h}_n\}$, which is defined as the sequence of distinct elements in

$$\left\{\ldots, \frac{E_k - l_k}{2}, \frac{E_k - l_k}{2} + 1, \ldots \frac{E_k + l_k}{2} - 1, \frac{E_k + l_k}{2}, \frac{E_{k+1} - l_{k+1}}{2}, \ldots\right\}, \tag{4}$$

and $m$ then labels the different occurrences of a given $\bar{h}_n$. This provides a bijection from the terms in the sum in equation (1) to those in the sums in equation (3). The coefficients $d_{m,n}$ are again non-negative.

(The sequence $\{\bar{h}_n\}$ is not monotonically increasing and is expected to contain accumulation points. Also, by the structure of the conformal blocks we know that the sum over $m$ in each $f_n(w)$ is infinite unless $n = 0$. We therefore need Fubini's theorem to justify the reshuffling in terms of the two infinite sums in equation (3).)

The main function of interest will now be:

$$F_\Lambda(w) := \Lambda^{-\Delta} \Gamma(\Delta + 1) \sum_{\{n : \bar{h}_n \le \Lambda\}} f_n(w). \tag{5}$$

Note that this restricted sum generally still contains infinitely many terms.

**Proposition 3.1.** *For fixed $\Lambda$, $F_\Lambda(w)$ is finite and analytic in $w$ if $\operatorname{Re}(w) < 0$.*

*Proof.* Consider first fixed real $w < 0$. Here each term in $f_n(w)$ and therefore in $F_\Lambda(w)$ is positive, and the full sum converges because

$$\sum_{\{n : \bar{h}_n \le \Lambda\}} f_n(w) \le \sum_{\{n : \bar{h}_n \le \Lambda\}} f_n(w)(1/2)^{\bar{h}_n - \Lambda} \le \sum_n f_n(w)(1/2)^{\bar{h}_n - \Lambda} = 2^{\Lambda - \Delta} g(w, 1/2). \tag{6}$$

The full sum is then also absolutely convergent for any $w$ in the left half plane, because $|e^{(h_{m,n} - \Delta)w}| = e^{(h_{m,n} - \Delta)\operatorname{Re}(w)}$ and therefore $|f_n(w)| \le f_n(\operatorname{Re}(w))$. Furthermore, on compact subsets the convergence of the sum is uniform because it is never slower than at the rightmost point in the subset. Therefore, $F_\Lambda(w)$ is analytic in $w$. $\qquad\square$

Now as $\bar{z} \to 1$ the behavior of $G(z, \bar{z})$ is dominated by the aforementioned identity contribution:

$$G(z, \bar{z}) = (1 - \bar{z})^{-\Delta} \left((1 - z)^{-\Delta} + O((1 - \bar{z})^{\hat{h}})\right), \tag{7}$$

with $\hat{h} := \inf_{n>0} \bar{h}_n > 0$ by the twist gap assumption. This expression uses crossing symmetry so it holds pointwise for any $z$ as long as $0 < |1 - z| < 1$.

**Proposition 3.2.** *Pointwise for any real $w < 0$,*

$$\lim_{\Lambda \to \infty} F_\Lambda(w) = (1 - e^w)^{-\Delta}. \tag{8}$$

*Proof.* For fixed real $w < 0$ both (7) and (3) are valid, and of course $g(w, \bar{z}) = G(e^w, \bar{z})$. In the same region $F_\Lambda(w)$ is positive and non-decreasing as a function of $\Lambda$, so we can define integrals using $dF_\Lambda$. The proposition is therefore a direct consequence of the Hardy-Littlewood Tauberian theorem A.1 as stated in the appendix. $\qquad\square$

**Theorem 3.3.** *For any $w$ with $\operatorname{Re}(w) < 0$,*

$$\lim_{\Lambda \to \infty} F_\Lambda(w) = (1 - e^w)^{-\Delta}, \tag{9}$$

*with the limit being uniform on compact subsets.*

*Proof.* Consider a compact subset $K$ such that $\text{Re}(w) \leq \epsilon < 0$ for all $w \in K$. Without loss of generality we will take $K$ to be connected and contain a segment of the real line. Then within $K$ we can use

$$|e^{w\Delta} F_\Lambda(w)| \leq e^{\text{Re}(w)\Delta} F_\Lambda(\text{Re}(w)) \leq e^{\epsilon\Delta} F_\Lambda(\epsilon), \tag{10}$$

where the last inequality uses that $e^{w\Delta} F_\Lambda(w)$ is monotonically increasing along the negative real axis. Furthermore, for sufficiently large $\Lambda$ we know that $e^{\epsilon\Delta} F_\Lambda(\epsilon) < C_\epsilon$ for some $\Lambda$-independent constant $C_\epsilon$ since otherwise the limit of proposition 3.2 would not exist for $w = \epsilon$. Therefore on $K$ we know that $F_\Lambda(w)$ is bounded uniformly in both $\Lambda$ and $w$. But then our claim follows immediately from proposition 3.2 and Vitali's convergence theorem A.4. $\qquad\square$

## 4 Consequences

**Monomial twist density**

Consider $\phi \in \mathcal{D}(\mathbb{R})$, a compactly supported infinitely smooth test function. We define the *monomial twist density*, a distribution $H_\Lambda \in \mathcal{D}'(\mathbb{R})$ as

$$\int d\eta\, \phi(\eta) H_\Lambda(\eta) := \frac{e^{-\Delta}\Lambda^\Delta}{\Gamma(\Delta+1)} \int dy \left[ \int \frac{d\eta}{2\pi} \phi(\eta) e^{\eta-iy\eta} \right] e^{i\Delta y} F_\Lambda(-1+iy). \tag{11}$$

The inner integral is the Fourier transform of an element of $\mathcal{D}(\mathbb{R})$ and therefore falls off faster than any power at large $|y|$. The outer integral is then also well-defined, because $F_\Lambda(-1+iy)$ as a function of $y \in \mathbb{R}$ is both smooth (by proposition 3.1) and bounded (by equation (10)).

The significance of the monomial twist density is that it can be written as:

$$H_\Lambda(\eta) = \sum_{\{n : \bar{h}_n \leq \Lambda\}} \sum_m d_{m,n} \delta(\eta - h_{m,n}). \tag{12}$$

Note that there are finitely many non-zero terms in the sum with support below any fixed value of $\eta$, since $E = h_{m,n} + \bar{h}_n \leq \eta + \Lambda$ and we assumed that $\{E_k\}$ is discrete. Our main claim is now that

$$\lim_{\Lambda \to \infty} \Lambda^{-\Delta} \int d\eta\, \phi(\eta) H_\Lambda(\eta) = \sum_{m=0}^\infty \frac{\Gamma(m+\Delta)}{m!\,\Gamma(\Delta)\Gamma(\Delta+1)} \phi(\Delta+m). \tag{13}$$

Indeed, the dominated convergence theorem justifies swapping the limit with the integral over $y$, and the rest is calculus. The right-hand side of equation (13), when multiplied by $\Lambda^\Delta$, describes the asymptotic behavior of the monomial twist density for $G(z,\bar{z}) = (1-z)^{-\Delta}(1-\bar{z})^{-\Delta}$. This leads to our main conclusion, valid for any function obeying the properties given in section 2: as an element of $\mathcal{D}'(\mathbb{R})$, the monomial twist density becomes that of the generalized free field to leading order at large $\Lambda$.

Suppose now that $\phi(\eta)$ is compactly supported in the interval $[\Delta + k - \epsilon, \Delta + k + \epsilon]$ with $k \in \mathbb{N}_0$ and $\epsilon > 0$. The right-hand side of equation (13) is finite, but the left-hand side goes to zero unless there exist infinitely many monomials $z^{h_{m,n}} \bar{z}^{\bar{h}_n}$ for which $h_{m,n}$ lies in that interval. The sum of the coefficients of these terms must furthermore behave as dictated by the generalized free field. This establishes the existence of the so-called double-twist families of monomials in $z$ and $\bar{z}$.

In appendix B we use the structure of conformal blocks and a Tauberian theorem of [15] to extend this result from monomials to conformal blocks.

**Lightcone limit on the second sheet**

Theorem 3.3 controls the asymptotics of $F_\Lambda(w)$ for any $w$ with $\mathrm{Re}(w) < 0$. In this region we can then employ the Abelian theorem A.2 to deduce that:

$$\lim_{\bar{z} \nearrow 1}(1 - \bar{z})^\Delta g(w, \bar{z}) = (1 - e^w)^{-\Delta}. \tag{14}$$

This limit holds uniformly on compact subsets, and is now seen to apply far beyond the original domain of validity of equation (7). Equation (14) directly constrains the lightcone limit on the second sheet. For example, if we define

$$\mathrm{dDisc}_s g(w, \bar{z}) := g(w, \bar{z}) - \frac{1}{2}\left(g(w + 2\pi i, \bar{z}) + g(w - 2\pi i, \bar{z})\right), \tag{15}$$

then we can conclude that

$$\lim_{\bar{z} \nearrow 1}(1 - \bar{z})^\Delta \mathrm{dDisc}_s g(w, \bar{z}) = 0. \tag{16}$$

In the future we plan to use this result as a stepping stone towards proving several of the claims in the works mentioned in the introduction.

Equation (14) can also be obtained more straightforwardly by considering $(1 - \bar{z})^\Delta g(w, \bar{z})$ as a family of functions labeled by $0 < \bar{z} < 1$. For $\mathrm{Re}(w) < 0$ each member of the family is holomorphic and as a whole the family is locally bounded (as one may easily verify). Therefore Vitali's theorem applies again, and we can infer the validity of equation (14) everywhere in the left half $w$ plane because it holds along the negative real $w$ axis. This simpler proof however only works at the leading order, and for subleading terms the use of the Abelian theorem seems unavoidable.

# 5 Outlook

It will be interesting to understand the size of the corrections to the results presented here. The best estimates will likely utilize our knowledge of the complex $\bar{z}$ plane and rely on one of the Tauberian theorems with remainder proven by Subhankulov [21].[2] We imagine this will allow us to estimate corrections for the behavior of $F_\Lambda(w)$ and $H_\Lambda(\eta)$ as described in theorem A.2 and in equation (13). We would also like to investigate whether equation (13) can be useful for proving dispersion relations in Mellin space [11, 12].

# Acknowledgments

The author is grateful to Petr Kravchuk, Dalimil Mazac, Thomas Pochart, Slava Rychkov and especially Jiaxin Qiao for valuable comments on the draft.

**Funding information** He acknowledges funding from the European Union (ERC "QFTinAdS", project number 101087025).

# A Mathematical results

The following is almost literally theorem I.15.3 from [23], to which we refer for a proof.

---

[2]See [22] for a translation of some theorems and the first application of these results to conformal correlators.

**Theorem A.1** (Hardy-Littlewood Tauberian theorem for the Laplace transform). *Let $F(\Lambda)$ vanish for $\Lambda < 0$, be non-decreasing, continuous from the right and such that the Stieltjes integral*

$$g(\bar{z}) = \int_{0-}^{\infty} \bar{z}^v \, dF(v), \tag{A.1}$$

*exists for $0 < \bar{z} < 1$. (The lower bound $0-$ means $\epsilon \nearrow 0$.) Suppose that for some constant $\Delta > 0$,*

$$\lim_{\bar{z} \nearrow 1} (1 - \bar{z})^{\Delta} g(\bar{z}) = 1. \tag{A.2}$$

*Then*

$$\lim_{\Lambda \to \infty} \Lambda^{-\Delta} \Gamma(\Delta + 1) F(\Lambda) = 1. \tag{A.3}$$

Tauberian theorems use the behavior of the integral to deduce a property of the integrand. Results in the opposite direction are called Abelian theorems, and they are easier to prove. In the main text we use the following theorem. Note that $F(\Lambda)$ is no longer required to be monotonic.

**Theorem A.2.** *Let $F(\Lambda)$ vanish for $\Lambda < 0$, continuous from the right and of bounded variation such that the Stieltjes integral*

$$g(\bar{z}) = \int_{0-}^{\infty} \bar{z}^v \, dF(v), \tag{A.4}$$

*exists for $0 < \bar{z} < 1$. If $F(\Lambda)$ furthermore obeys equation (A.3), then $g(\bar{z})$ obeys equation (A.2).*

*Proof.* We set $\bar{z} = e^{-t}$ with $t > 0$ and will show that $\lim_{t \searrow 0} t^{\Delta} g(e^{-t}) = 1$. Using the ability [23] to integrate by parts we can write

$$t^{\Delta} g(e^{-t}) - 1 = t^{\Delta} \int_{0-}^{\infty} e^{-vt} dF(v) - 1 = t^{\Delta+1} \int_{0}^{\infty} \left( F(v) - \frac{v^{\Delta}}{\Gamma(\Delta+1)} \right) e^{-vt} dv. \tag{A.5}$$

Equation (A.3) says that for any $\epsilon > 0$ there exists a $v_* > 0$ such that $|v^{-\Delta}\Gamma(\Delta+1)F(v) - 1| \le \epsilon/2$ for all $v \ge v_*$. Therefore

$$\begin{aligned}
|t^{\Delta} g(e^{-t}) - 1| &\le t^{\Delta+1} \int_{0}^{v_*} \left| F(v) - \frac{v^{\Delta}}{\Gamma(\Delta+1)} \right| e^{-vt} dv + \frac{\epsilon}{2} \times t^{\Delta+1} \int_{v_*}^{\infty} \frac{v^{\Delta}}{\Gamma(\Delta+1)} e^{-vt} dv \\
&\le t^{\Delta+1} \int_{0}^{v_*} \left| F(v) - \frac{v^{\Delta}}{\Gamma(\Delta+1)} \right| dv + \frac{\epsilon}{2}.
\end{aligned} \tag{A.6}$$

The remaining integral is finite and $t$-independent, so the expression on the last line is less than $\epsilon$ for sufficiently small $t$. We conclude that the $t \searrow 0$ limit of the left-hand side is zero. $\square$

**Theorem A.3** (Montel's theorem). *Let $\Omega$ be a domain in $\mathbb{C}$. Consider a family $\mathcal{F}$ of holomorphic functions on $\Omega$. We suppose that $\mathcal{F}$ is locally bounded, meaning that for every $z \in \Omega$, there is an open set $U \subset \Omega$ and $M > 0$ such that $|f(z)| \le M$ for all $z \in U$ and all $f \in \mathcal{F}$. Then every sequence in $\mathcal{F}$ contains a subsequence that converges uniformly on compact subsets to a holomorphic function.*

Montel's theorem is a fundamental result in complex analysis, see for example [24]. It allows for a straightforward proof of Vitali's theorem. Below we give both theorem and proof from [25], where it is referred to as the Vitali-Porter theorem.

**Theorem A.4** (Vitali's theorem)**.** *Let $\{f_n\}$ be a locally bounded sequence of analytic functions in a domain $\Omega$ such that $\lim_{n\to\infty} f_n(z)$ exists for each $z$ belonging to a set $E \subseteq \Omega$ which has an accumulation point in $\Omega$. Then $\{f_n\}$ converges uniformly on compact subsets of $\Omega$ to an analytic function.*

*Proof.* By Montel's theorem our sequence contains a uniformly convergent subsequence, defining some function $f(z)$ on $\Omega$. Without loss of generality we may assume that $f(z) = 0$. Now suppose the theorem is false, so $\lim_{n\to\infty} f_n(z)$ does not exist or is not uniform on some compact subset. It must then be possible to find an $\epsilon > 0$, a compact subset $K$ of $\Omega$, and another subsequence $\{f_k\}$ such that $\sup_{z \in K} |f_k(z)| \geq \epsilon$ for all $k$. Furthermore, the compact subset $K$ can always be chosen to include an open neighborhood of the accumulation point of $E$. But such a subsequence must have itself a uniformly convergent subsubsequence, again by Montel's theorem. On $E$ the subsubsequence converges to 0, but since $\epsilon > 0$ the limiting function cannot be identically zero. This contradicts the identity theorem, which says [24] that the zeroes of a non-identically zero analytic function are isolated. $\qquad\square$

# B  Lightcone and conformal blocks

In this appendix we will strengthen the results in the main text from monomials $z^h \bar{z}^{\bar{h}}$ to full conformal blocks. We first need to discuss an intermediate result for lightcone conformal blocks, based on a Tauberian theorem from [15].

## B.1  Lightcone blocks

For $\nu \geq 0$ and $0 < \bar{z} < 1$ we define the *lightcone conformal block* as:

$$\kappa(\nu, \bar{z}) := \frac{(\bar{z}/4)^{\nu}}{\sqrt{\nu+1}} {}_2F_1(\nu, \nu, 2\nu, \bar{z}). \tag{B.1}$$

**Theorem B.1** (Tauberian theorem for the conformal bootstrap [15])**.** *Let $F(\Lambda)$ vanish for $\Lambda < 0$, be nondecreasing, continuous form the right and such that the Stieltjes integral*

$$g(\bar{z}) = \int_{0-}^{\infty} \kappa(\nu, \bar{z}) dF(\nu), \tag{B.2}$$

*exists for $0 < \bar{z} < 1$. Suppose that for some constant $\Delta > 0$,*

$$\lim_{\bar{z} \nearrow 1} (1 - \bar{z})^{\Delta} g(\bar{z}) = 1. \tag{B.3}$$

*Then*

$$\lim_{\Lambda \to \infty} \Lambda^{-2\Delta} F(\Lambda) = \frac{2\sqrt{\pi}}{\Gamma(\Delta)\Gamma(\Delta+1)}. \tag{B.4}$$

For the proof we refer to [15]. We note the unconventional prefactor $4^{-\nu}/\sqrt{\nu+1}$ in equation (B.1), which arises from a Bessel function approximation to the lightcone conformal blocks at large $\nu$.[3]

Our analysis begins with the substitution of $\kappa(\bar{h}, \bar{z})$ instead of the monomials $\bar{z}^{\bar{h}}$ in equation (3). This produces a decomposition in lightcone conformal blocks:

$$g(w, \bar{z}) = \bar{z}^{-\Delta} \sum_n f_n^{(\mathrm{lc})}(w) \kappa(\bar{h}_n, \bar{z}), \qquad f_n^{(\mathrm{lc})}(w) := \sum_m d_{m,n}^{(\mathrm{lc})} e^{(h_{m,n} - \Delta)w}. \tag{B.5}$$

---

[3]This unconventional prefactor was $4^{-\nu}/\sqrt{\nu}$ in [15]. We changed it in order to avoid notational hassles if $F(0) \neq 0$. This change does not affect the claimed result since the two prefactors are smooth and become equal for large $\nu$.

Any function with the properties described in section 2 admits such a decomposition, but the CFT axioms also impose that the coefficients $d_{m,n}^{(\mathrm{lc})}$ should again be non-negative, which is a non-trivial constraint.

If we now define

$$F_\Lambda^{(\mathrm{lc})}(w) := \frac{1}{2\sqrt{\pi}} \Lambda^{-2\Delta} \Gamma(\Delta)\Gamma(\Delta+1) \sum_{\{n\,:\,\bar{h}_n \leq \Lambda\}} f_n^{(\mathrm{lc})}(w), \tag{B.6}$$

then we claim that, for all $w$ in the left half plane,

$$\lim_{\Lambda \to \infty} F_\Lambda^{(\mathrm{lc})}(w) = (1 - e^w)^{-\Delta}. \tag{B.7}$$

The argument is the same as before: pointwise for real $w < 0$ we use theorem B.1 and for general complex $w$ we can use Vitali's theorem. Taking the Fourier transform then implies that the density

$$H_\Lambda^{(\mathrm{lc})}(\eta) := \sum_{\{n\,:\,\bar{h}_n \leq \Lambda\}} \sum_m d_{m,n}^{(\mathrm{lc})} \delta(\eta - h_{m,n}), \tag{B.8}$$

converges to the generalized free field value: for any $\phi \in \mathcal{D}(\mathbb{R})$,

$$\lim_{\Lambda \to \infty} \Lambda^{-2\Delta} \int d\eta\, \phi(\eta) H_\Lambda^{(\mathrm{lc})}(\eta) = \sum_{m=0}^{\infty} \frac{2\sqrt{\pi}\Gamma(m+\Delta)}{m!\Gamma(\Delta)^2\Gamma(\Delta+1)} \phi(\Delta+m). \tag{B.9}$$

There must therefore be, for any $k \in \mathbb{N}_0$ and $\epsilon > 0$, infinitely many lightcone conformal blocks in the decomposition (B.5) for which the corresponding factor of $z^h$ lies in the interval $[\Delta + k - \epsilon, \Delta + k + \epsilon]$.

## B.2 Conformal blocks

The functions $G(z,\bar{z})$ that concern us also admit another decomposition, into conformal blocks $\mathcal{G}_{h,\bar{h}}(z,\bar{z})$:

$$G(z,\bar{z}) = z^{-\Delta}\bar{z}^{-\Delta} \sum_q d_q^{(\mathrm{p})} \mathcal{G}_{h_q,\bar{h}_q}(z,\bar{z}), \tag{B.10}$$

with coefficients $d_q^{(\mathrm{p})}$ which are again non-negative by the CFT axioms.

There are many ways to define conformal blocks. We will use the results of [20] and write them as:

$$\mathcal{G}_{h,\bar{h}}(z,\bar{z}) = \sum_{n=0}^{\infty} z^{h+n} \sum_{m=-n}^{n} \alpha_{n,m}(h,\bar{h}) \kappa(\bar{h}+m,\bar{z}). \tag{B.11}$$

The sums are again absolutely convergent for $0 < |z|, |\bar{z}| < 1$. The labels $h$ and $\bar{h}$ are the powers of the leading monomial, $z^h \bar{z}^{\bar{h}}$, in the double limit where we first send $z \to 0$ and then $\bar{z} \to 0$. The term with $q = 0$ in equation (B.10) has $h_0 = \bar{h}_0 = 0$ and $d_0^{(\mathrm{p})} = 1$ and corresponds to the identity conformal block, $\mathcal{G}_{0,0}(z,\bar{z}) = 1$. For $q > 0$ we assume that all the $h_q$ and $\bar{h}_q$ are compatible with the axioms for a unitary CFT with a twist gap, which in particular implies that $\bar{h}_q - h_q \in \mathbb{N}_0$ and $h_q > \hat{h}$ for some $\hat{h} > 0$.

Note that conformal blocks are symmetric, $\mathcal{G}_{h,\bar{h}}(z,\bar{z}) = \mathcal{G}_{h,\bar{h}}(\bar{z},z)$, and therefore $\alpha_{n,m}(h,\bar{h}) = 0$ for $m < h - \bar{h}$. The other coefficients can be determined from a recursion relation [26] up to an overall normalization, which we fix by choosing $\alpha_{0,0}(h,\bar{h}) = 1$. They are then rational functions modulo a square root that arises from the unconventional prefactor in equation (B.1). Importantly, the $\alpha_{m,n}(h,\bar{h})$ remain finite as $\bar{h} \to \infty$ and we can write:

$$\alpha_{n,m}(h,\bar{h}) = \alpha_{n,m}(h,\infty) + \delta\alpha_{n,m}(h,\bar{h}), \tag{B.12}$$

with $\alpha_{n,m}(h,\infty) \geq 0$ and with

$$|\delta\alpha_{n,m}(h,\bar{h})| < \frac{C_{n,m}(h)}{1+\bar{h}}. \tag{B.13}$$

Below we will need that $h \mapsto C_{n,m}(h)$ can be taken continuous and that $h \mapsto \alpha_{n,m}(h,\infty)$ is smooth.[4]

Given the conformal block decomposition in equation (B.10), we define the *primary twist density* as:

$$H_\Lambda^{(p)}(\eta) = \sum_{\{q:\bar{h}_q \leq \Lambda\}} d_q^{(p)} \delta(\eta - h_q). \tag{B.14}$$

For finite $\Lambda$ this distribution has a well-defined action on any compactly supported test function. In $\mathcal{D}'(\mathbb{R})$ the following theorem shows that the large $\Lambda$ limit is universal.

**Theorem B.2.** *Given the above properties of conformal blocks,*

$$\lim_{\Lambda\to\infty} \Lambda^{-2\Delta} \int d\eta\, \phi(\eta) H_\Lambda^{(p)}(\eta), \tag{B.15}$$

*is finite for any $\phi \in \mathcal{D}(\mathbb{R})$ and completely determined by equation (B.9).*

It will be useful to introduce

$$\mathcal{D}(\eta_*) := \mathcal{D}((-\infty, \eta_*)), \tag{B.16}$$

which is the space of smooth test functions whose support is a compact subset of $(-\infty, \eta_*)$.

*Proof.* The structure of the conformal block given in equation (B.11) and the decomposition (B.12) of the coefficients $\alpha_{n,m}(h,\bar{h})$ allow us to write:

$$H_\Lambda^{(p)}(\eta) = H_\Lambda^{(lc)}(\eta) - \delta_I H_\Lambda^{(p)}(\eta) - \delta_{II} H_\Lambda^{(p)}(\eta), \tag{B.17}$$

with

$$\delta_I H_\Lambda^{(p)}(\eta) = \sum_{n=1}^\infty \sum_{m=-n}^n \sum_{\{q:\bar{h}_q \leq \Lambda-m\}} d_q^{(p)} \alpha_{n,m}(h_q,\infty)\delta(\eta - h_q - n),$$

$$\delta_{II} H_\Lambda^{(p)}(\eta) = \sum_{n=1}^\infty \sum_{m=-n}^n \sum_{\{q:\bar{h}_q \leq \Lambda-m\}} d_q^{(p)} \delta\alpha_{n,m}(h_q,\bar{h}_q)\delta(\eta - h_q - n). \tag{B.18}$$

Only finitely many terms of the infinite sums survive whenever $\delta_I H_\Lambda^{(p)}$ or $\delta_{II} H_\Lambda^{(p)}$ act on a compactly supported test function. To see this, suppose that $\phi \in \mathcal{D}(\eta_*)$ and recall that $h_q, \bar{h}_q \geq 0$. Then all the non-zero terms must obey

$$1 \leq n < \eta_*, \qquad 0 \leq h_q + \bar{h}_q < \Lambda + \eta_*. \tag{B.19}$$

These are finite in number since $\lim_{q\to\infty} h_q + \bar{h}_q = \infty$ by the discrete spectrum assumption.

Our proof will proceed by induction. Equation (B.19) implies that $\delta_I H_\Lambda^{(p)}(\eta)$ and $\delta_{II} H_\Lambda^{(p)}(\eta)$ do not contribute if $\eta_* \leq 1$. So for this subspace of test functions the theorem is trivial:

$$\forall \phi \in \mathcal{D}(1), \qquad \lim_{\Lambda\to\infty} \Lambda^{-2\Delta} \int d\eta\, \phi(\eta) H_\Lambda^{(p)}(\eta) = \lim_{\Lambda\to\infty} \Lambda^{-2\Delta} \int d\eta\, \phi(\eta) H_\Lambda^{(lc)}(\eta). \tag{B.20}$$

---

[4]One way to see that the $\alpha_{n,m}(h,\bar{h})$ do not diverge as $\bar{h} \to \infty$ is because that would be inconsistent with what follows.

We will now adapt this result to $\phi \in \mathcal{D}(\eta_*)$ for any real $\eta_*$, by increasing $\eta_*$ in steps of size $1/2$.

Suppose then that that equation (B.15) is known for all test functions in $\mathcal{D}(\eta_*)$. To obtain the result for $\phi \in \mathcal{D}(\eta_* + 1/2)$ we apply both sides of equation (B.17) to such a test function. The right-hand side contains three terms. The first term is the lightcone twist density for which the large $\Lambda$ behavior is given in equation (B.9). The second term yields

$$\int d\eta\, \phi(\eta)\delta_I H_\Lambda^{(p)}(\eta) = \sum_{n=1}^{\lfloor \eta_*+1/2 \rfloor} \sum_{m=-n}^{n} \int d\eta'\, \alpha_{n,m}(\eta', \infty)\phi(\eta'+n)H_{\Lambda-m}^{(p)}(\eta').$$
(B.21)

The large $\Lambda$ behavior of each summand on the right-hand side is known by the induction hypothesis, since the maps $\eta \mapsto \alpha_{n,m}(\eta, \infty)\phi(\eta+n)$ are elements of $\mathcal{D}(\eta_*)$.

This leaves us with the last term on the right-hand side of equation (B.17). We know that the contribution of $\delta_{II} H_\Lambda^{(p)}(\eta)$ vanishes for $\phi \in \mathcal{D}(1)$, and the next lemma then implies that it remains subleading for any $\phi \in \mathcal{D}(\mathbb{R})$. This establishes the theorem. $\qquad\square$

**Lemma B.3.** *Let $\eta_* > 0$ and $\Delta > 0$ and suppose that for all $\chi \in \mathcal{D}(\eta_*)$,*

$$\lim_{\Lambda \to \infty} \Lambda^{-2\Delta} \int d\eta\, \chi(\eta) H_\Lambda^{(p)}(\eta) < \infty.$$
(B.22)

*Then for all $\phi \in \mathcal{D}(\eta_* + 1/2)$,*

$$\lim_{\Lambda \to \infty} \Lambda^{-2\Delta} \int d\eta\, \phi(\eta)\delta_{II} H_\Lambda^{(p)}(\eta) = 0.$$
(B.23)

*Proof.* Let $\phi \in \mathcal{D}(\eta_* + 1/2)$ and consider:

$$\int d\eta\, \phi(\eta)\delta_{II} H_\Lambda^{(p)}(\eta) = \sum_{n=1}^{\lfloor \eta_*+1/2 \rfloor} \sum_{m=-n}^{n} \sum_{\{q\,:\,\bar{h}_q+m\leq\Lambda\}} d_q^{(p)}\, \delta\alpha_{n,m}(h_q, \bar{h}_q)\, \phi(h_q+n).$$
(B.24)

Since only finitely many terms contribute, we can immediately write:

$$\left| \int d\eta\, \phi(\eta)\delta_{II} H_\Lambda^{(p)}(\eta) \right| \leq \sum_{n=1}^{\lfloor \eta_*+1/2 \rfloor} \sum_{m=-n}^{n} \sum_{\{q\,:\,\bar{h}_q+m\leq\Lambda\}} \frac{d_q^{(p)}}{1+\bar{h}_q} C_{n,m}(h_q)|\phi(h_q+n)|.$$
(B.25)

For fixed $n$ and $m$ consider now the function $h \mapsto C_{n,m}(h)|\phi(h+n)|$. We will replace this non-negative continuous function, compactly supported below $\eta_* - 1/2$, with a non-negative test function $\chi_{n,m} \in \mathcal{D}(\eta_*)$ such that, for all $\eta \in \mathbb{R}$,

$$C_{n,m}(\eta)|\phi(\eta+n)| \leq \chi_{n,m}(\eta).$$
(B.26)

Equation (B.25) is then bounded from above by a finite sum of terms of the form:

$$\int d\eta\, \chi(\eta) \sum_{\{q\,:\,\bar{h}_q\leq\Lambda-m\}} \frac{d_q^{(p)}}{1+\bar{h}_q}\delta(\eta-h_q),$$
(B.27)

for non-negative real $\chi \in \mathcal{D}(\eta_*)$. Splitting the sum at $\sqrt{\Lambda}$, we can further bound this as:

$$\int d\eta\, \chi(\eta) \sum_{\{q\,:\,\bar{h}_q\leq\Lambda\}} \frac{d_q^{(p)}}{1+\bar{h}_q}\delta(\eta-h_q) \leq \int d\eta\, \chi(\eta) H_{\sqrt{\Lambda}}^{(p)}(\eta) + \frac{1}{1+\sqrt{\Lambda}} \int d\eta\, \chi(\eta) H_\Lambda^{(p)}(\eta).$$
(B.28)

But when multiplied with $\Lambda^{-2\Delta}$ both terms vanish as $\Lambda \to \infty$, by our hypothesis (B.22). $\qquad\square$

In the generalized free field theory the coefficients $d_q^{(p)}$ and spectrum $(h_q, \bar{h}_q)$ are explicitly known, see equation (43) in [27]. The large $\Lambda$ limit of the primary twist density is given by:

$$\lim_{\Lambda \to \infty} \Lambda^{-2\Delta} \int d\eta \, \phi(\eta) H_\Lambda^{(p)}(\eta) = \sum_{n=0}^{\infty} \frac{2\sqrt{\pi}\Gamma(\Delta - \frac{d}{2} + 1 + n)^2 \Gamma(2\Delta - d + 2n + 1)}{n! \Gamma(\Delta)\Gamma(\Delta + 1)\Gamma(\Delta - \frac{d}{2} + 1)^2 \Gamma(2\Delta - d + n + 1)} \phi(\Delta + n). \quad \text{(B.29)}$$

Our theorem states that this answer is universal and thus must hold for any function satisfying the assumptions spelled out in section 2 and with a conformal block decomposition with positive coefficients.

Consider once more $\phi$ compactly supported in the interval $[\Delta + k - \epsilon, \Delta + k + \epsilon]$ for $k \in \mathbb{N}_0$ and $\epsilon > 0$. If $\Delta > d/2 - 1$ and $\phi(\Delta + k) \neq 0$ then the right-hand side of equation (B.29) is not zero, so the primary twist density acting on such a $\phi$ diverges like $\Lambda^{2\Delta}$. This can only happen if there are infinitely many conformal blocks $\mathcal{G}_{h,\bar{h}}(z, \bar{z})$ with $h$ in this interval.

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
