# Peer review of "Theorems for the Lightcone Bootstrap"

_SciPost Physics, doi:SciPost Phys. 18, 207 (2025)_

## Round 1 · Referee Report · Sridip Pal (Referee 1) · 2025-4-15

Strengths

  1. Lightcone conformal bootstrap was originally formulated in 2012 in Ref [1,2] and has widely been successful. However, a mathemetically rigorous understanding of the original arguments has been missing and long sought-after. In recent years, progress has been made in this direction such as by Ref [18] in 2022. The present paper goes way beyond that and proves all the essential claims of the lightcone conformal bootstrap. Thus, in my opinion, the core strength of this paper is to solve a long-standing challenging problem in a mathematically rigorous way.

  2. The paper has a clear potential for widely varied applications and extensions. In fact, one can apply the technique (or a refined version of it) in the context of non-rational 2D CFTs and find universal results in the large spin sector.

  3. It is clearly written in spite of being technically involved.

Weaknesses

None that I can see.

Report

This paper not only meets the journal's acceptance criteria but exceeds them. I strongly recommend it for publication. Please refer to the 'Strengths' section above for further details.

I have some minor suggestions that may improve clarity, but I mostly leave it to the author to decide whether to implement them.

  1. In Eq. 4) I would suggest writing $\frac{E_l+l_k}{2}-1$ as $\frac{E_k-l_k}{2}+(l_k-1)$ and $\frac{E_l+l_k}{2}$ as $\frac{E_k-l_k}{2}+l_k$ to make the pattern transparent.

  2. In the last line of Proposition 3.1, the author presumably invoked Weiestrass' theorem from complex analysis to deduce the analyticity. It would be helpful to mention this explicitly, as it provides a useful keyword for readers unfamiliar with the technique.

  3. The fact that the limit in Theorem 3.3 is uniform on compact subsets does not appear to be used in later arguments. In Section 4, boundedness (Eq 10) is required for the application of DCT, and the existence of the limit is used to deduce Eq. (13). If uniformity plays a role subtly, it would be worth pointing out. If not, a remark to that effect could help reassure readers that they are not missing a delicate point.

  4. It might be nice to give an example of a compactly supported infinitely smooth test function in section 4.

  5. I would recommend moving Appendix B to the main text. I understand the key mathematical ideas are already there in the main text. However, the results involving primaries are more exciting and hence deserve a place in the main text in my opinion.

  6. The last line of section 4 is mysterious to me. I would have thought that it is easier to access subleading information at the level of correlator compared to the level of density. This is indeed the case, for example, in the context of finding universal results about the density of states in 2D CFT. It would be nice to clarify why the use of the Abelian theorem seems unavoidable to the author for the subleading terms.

Comments regarding Appendix B:

  1. Are equations 34, 35 proven, in particular, all the properties required such as smoothness and finiteness, positivity? It will be helpful for the readers to include the proof or refer to relevant papers.

  2. I would suggest putting in more words around Eq 40 to explain the strategy of the proof.

  3. I would suggest splitting Eq 41 into two parts since only the second inequality is needed to show that finitely many terms contribute. The other inequality $1\leqslant n<\eta_*$ is used later to deduce Eq 42.

  4. It would be nice to mention the following cross-check: one can take Eq. 51 and integrate against a test function from $\mathcal{D}(\eta_*)$ with $0<\Delta<\eta_*<1$ and verify Eq. 42 holds. This amounts to checking that the $m=0$ term in the R.H.S of Eq. 31 is same as the $n=0$ term in the RHS of Eq. 51, which is indeed the case.

Requested changes

Please see the report.

Recommendation

Publish (surpasses expectations and criteria for this Journal; among top 10%)

---

## Round 1 · Referee Report · Anonymous (Referee 2) · 2025-5-16

Strengths

Clearly and concisely written

Rigorous proof of a key result in conformal bootstrap

New technique with promising future directions

Weaknesses

none

Report

A rigorous proof such as the one given in this paper has been a longtime goal of the bootstrap program. The key new technique is the use of Vitali's theorem, as is stated clearly in the introduction, justifying the uniform convergence of correlator $G(z,\bar{z})$ in the lightcone limit to the Generalized Free Field correlator as an analytic function for $0<|z|<1$, which can then be used to extract the spectrum of twists contributing in this limit.

I have only some minor requests of clarifications, see below.

Requested changes

1) In equation (1), why is the index $j$ required, instead of absorbing it into the definition of $\ell_k$? Doing this would simplify the notation and some equations, eq (4). Alternatively, if the motivation is to make the connection to irreducible representations of (subgroups of) the conformal algebra indexed by $k$, perhaps this could be explained and stated explicitly.

2) I think a few extra comments would be useful to make the proof of theorem 3.3 easier to follow, as it is the key step of the paper. Specifically, why can $K$ be taken without loss of generality to contain a segment of the real line? What are the subsets $E$ and $\Omega$ in the application of Vitali's theorem as stated in A.4?

3) In the outlook, future work is proposed exploring the size of corrections to the current results. This seems to suggest that little is known about such corrections in the context of the conformal bootstrap, but there is a large body of work on them, why not mention some of these?

Recommendation

Publish (surpasses expectations and criteria for this Journal; among top 10%)

---

## Round 1 · Referee Report · Anonymous (Referee 3) · 2025-5-19

Strengths

  • The article provides the first complete proof of a long-sought result in conformal field theory.

  • It is likely to lead to new inroads on other related problems.

  • It is written in a to-the-point and clear style.

Report

The article's main achievement is a fully rigorous (by math standards) proof of the existence of double-twist operators with asymptotic twists $2\Delta_\phi+2n$ for all $n\geq 0$, in a general unitary CFT with a twist gap. This result was previously known for $n=0$. The present article introduces new ideas (combining a Tauberian theorem with Vitali's theorem) to circumvent a previous impasse.

I find the article both highly original and correct. Indeed, once the right set of ingredients for the proof is identified, the result falls into place very naturally. This is quite unusual in a field known for its high degree of technical sophistication. I am hopeful that similar ideas will prove useful for attacking related problems in CFT.

Recommendation

Publish (surpasses expectations and criteria for this Journal; among top 10%)

---

## Editorial Decision

published